# Psilocybin ameliorates neuropathic pain-like behaviour in mice and facilitates gabapentin-mediated analgesia
Tatum Askey[1,4], Daniel Allen-Ross[1,4], Daniil Luzyanin[1], Reena Lasrado[2], Gary Gilmour[2], Stephen P. Hunt [3], Francesco Tamagnini [1], Maqsood Ahmed[2], Gary J. Stephens[1] & Maria Maiarú [1] ✉

Chronic pain states remain challenging to control with current drug therapies. Here, we demonstrate that a single dose of psilocybin produces a sustained anti-nociceptive effect in chronic neuropathic pain models in male and female mice, mediated primarily by 5-HT$_{2A}$ receptors. Critically, psilocybin significantly potentiates the analgesic efficacy of gabapentin, a standard-of-care treatment, representing the first preclinical evidence that a psychedelic can serve as a pain-network primer for existing analgesics. This finding represents a novel therapeutic strategy with potential clinical application, particularly for the 30-50% of neuropathic pain patients who fail gabapentin monotherapy. Our data demonstrate that a single psilocybin injection produces sustained month-long changes that enhance gabapentin efficacy in a preclinical model of human pain. Together, these findings indicate that psilocybin both acutely enhances analgesia and induces lasting changes that amplify gabapentin efficacy weeks later. Such a translation is notable in chronic pain management, where most analgesics require chronic dosing and lose efficacy through tolerance. These findings establish psilocybin as a potential therapeutic addition for pain management by enabling longer-lasting changes in pain-processing networks and enhancing the utility of established treatments.

Chronic pain affects millions of people worldwide and presents a significant social and economic burden. Long-lasting pain negatively impacts quality of life[1] and is associated with substantial unmet clinical need, as current pharmacological management is often limited by poor tolerability and addiction potential[2,3]. People with chronic pain frequently develop affective comorbidities, including depression and anxiety, which further worsen clinical outcomes[4].

Psilocybin, a classic psychedelic, produces profound alterations in perception and cognition[5] through activation of multiple serotoninergic receptors, with 5-hydroxytryptamine 2 A (5-HT$_{2A}$) receptor signalling likely necessary for its psychedelic effects[6]. Recent clinical interest has centered on psilocybin's potential for treating major depression and related mood disorders[7], such conditions commonly co-occur with chronic pain. Notably, single-dose psilocybin treatment has been shown to produce sustained therapeutic benefits associated with long-term reorganization of intrinsic brain networks and resetting of maladaptive patterns of neural connectivity[8]. Very recent preclinical evidence has demonstrated that a single dose of psilocybin rapidly and sustainably reverses both neuropathic

and inflammatory pain-like states while simultaneously alleviating co-occurring anxiodepressive behaviours in rodent models, with effects mediated through normalized activity in the anterior cingulate cortex (ACC), reportedly via partial agonism at both 5-HT$_{2A}$ and 5-HT$_{1A}$ receptors[9].

Early evidence suggests psilocybin may alleviate treatment-resistant pain conditions, including phantom limb pain[10], migraine[11], and inflammatory pain states in rodent models[12]. Whilst understanding of the neuronal networks sustaining chronic pain remains incomplete[13], recent work demonstrates that maladaptive functional and structural connectivity changes precede the onset of chronic neuropathic pain[14]. This observation raises the possibility that psilocybin could 'unlock' or remodel pain-processing networks by fundamentally altering their organization[15]. Furthermore, given that single-dose psilocybin produces sustained network-level changes without the tolerance development associated with conventional analgesics, we hypothesized that psilocybin could enhance the efficacy of established pain medications. Here, we characterize psilocybin effects in the spared nerve injury (SNI) mouse model of pain and demonstrate that

[1]Department of Pharmacology, School of Pharmacy, University of Reading, Reading, UK. [2]Compass Pathfinder Ltd (a subsidiary of Compass Pathways PLC), London, UK. [3]Department of Cell and Developmental Biology, University College London, London, UK. [4]These authors contributed equally: Tatum Askey, Daniel Allen-Ross. ✉e-mail: m.maiaru@reading.ac.uk

psilocybin potentiates the anti-nociceptive response of gabapentin, a standard-of-care treatment for neuropathic pain.

## Results

### Psilocybin shows anti-nociceptive effects in male and female mice after injury

We induced a chronic pain-like status in both adult C57BL/6 J male and female mice. The SNI mouse model is a well-established preclinical model of neuropathic pain that involves partial transection of the peripheral nerves innervating the hind paw[16]. The SNI model is considered one of the best experimental models for neuropathic pain research because it accurately reproduces key clinical features[17]: it mimics traumatic nerve injuries seen in human conditions such as surgery, trauma, or diabetes, preserves some nerve fibers while selectively damaging others, and produces pain-like behaviours, mechanical allodynia and thermal hyperalgesia, that closely parallel human neuropathic pain. This translational relevance makes it an excellent model for studying pain mechanisms and analgesic responses.

The head-twitch response (HTR) is considered a rodent behavioural proxy of the human response to a psychedelic experience[18] and psilocybin administration has been shown to induce HTR in mice[19], with a marked increase in this behaviour observed at a standard dose of 1 mg/kg. In a dose-response curve, the HTR seen after a 1 mg/kg dose was comparable to that induced by a higher dose of psilocybin[19]; based on this, we selected doses of 0.3 mg/kg and 1 mg/kg of psilocybin in the SNI model to evaluate its effects across a range of different behavioural tests that measure both reflexive and affective responses to mechanical and thermal stimulation of the hind paw.

The experimental design is summarised in Fig. 1A. Male and female mice underwent SNI surgery and, when static mechanical hypersensitivity was fully developed (between day 12 and day 14), they received a single intra-peritoneal (i.p.) injection of psilocybin (1 mg/kg) or saline control (Fig. 1A). An increase in HTR was observed in the psilocybin treatment groups compared to saline control both in male and in female mice (Fig. 1B) confirming central nervous system exposure to the drug. Mechanical hypersensitivity was reduced in male (%MPE = 25.5) and female (% MPE = 27) mice treated with psilocybin and in male mice the effect lasted up to day 28 after injection (test day 45) (Fig. 1C). This effect lasted one week in female (test day 21). The single dose of psilocybin had no adverse effect on locomotor performance in SNI mice (Supplementary Fig 1). In male mice, preliminary data also indicate that psilocybin (1 mg/kg) was also able to reduce hypersensitivity to light brush that develops after peripheral nerve injury (Fig. 1D). To assess cold sensitivity, we used the thermal place preference test (TPP)[20] and recorded the total amount of time mice spent on the cold plate before (Bs) and after nerve injury (day 1 to day 30) (Fig. 1E). An overall trend toward an increase in the time spent on the cold plate was observed ($P = 0.085$), compared to saline-injected mice which reached significance by day 30 (Bonferroni corrections $P = 0.02$) (Fig. 1E) (psilocybin, 65.1 ± 23.4 s; saline, 18.9 s ± 3.6 s) Psilocybin (1 mg/kg) also had no adverse effect on locomotor performance in naïve male mice (Supplementary Fig 2A-B). Next, we analysed faecal output as a measure of stress in mice[21] after SNI for psilocybin (1 mg/kg) vs saline controls. Psilocybin treatment reduced faecal boli output in mice after SNI surgery (Supplementary Fig 3), this was also associated with increased body weight (Supplementary Fig 3). These data are consistent with the hypothesis that psilocybin reduces the stress that follows injury, this aspect warrants further investigation. To further characterise the temporal profile of psilocybin's effects on mechanical sensitivity, we performed a detailed time-course from 30 min to 24 h post-injection (Fig. 1F). No significant effect of psilocybin was observed at 30 min and 1-hour post-injection; however, psilocybin-treated mice showed robust and sustained reductions in mechanical sensitivity from 2 h onwards, which persisted throughout the 24-hour assessment period (MPE = 46%). By contrast, saline-treated mice did not show changes in mechanical sensitivity over the same timeframe. Because of this long-lasting effect of psilocybin on pain-like behaviours, we tested whether a pre-emptive injection of psilocybin would be able to prevent the development of mechanical hypersensitivity (Fig. 1G). Interestingly, statistical analysis

showed that an injection of psilocybin given 30 days before the SNI surgery, failed to prevent the development of mechanical hypersensitivity (Fig. 1H).

We next tested whether 5-HT$_{2A}$R activation is required for the anti-nociceptive effect of psilocybin. In mice, psilocybin-induced HTR is mediated by the 5-HT$_{2A}$Rs[18]. We therefore pre-injected male mice with the 5-HT$_{2A}$R antagonist volinanserin (0.032 mg/kg i.p.), shown to be effective in previous mouse behavioural studies[22,23], followed 30 mins later by psilocybin (1 mg/kg i.p.) (Fig. 1I). Control mice received saline prior to psilocybin. Volinanserin pretreatment blocked the HTR (Fig. 1J) and substantially reduced its anti-nociceptive effect on mechanical hypersensitivity (Fig. 1K) and had no adverse effect on locomotory functions (Fig. 1L). Volinanserin alone has no effect on mechanical hypersensitivity (Supplementary Fig 4A). We also assessed the effect of psilocybin and volinanserin treatments on spontaneous behaviour in naïve mice (Supplementary Fig 4B–D). No differences were observed in total distance travelled across groups, supporting that neither psilocybin nor volinanserin impaired locomotor functions; these data also suggest that these behaviours may be mediated by a 5-HT$_{2A}$R independent mechanism[24]. Unsupported rearing and grooming behaviours were reduced by psilocybin 10 min post-injection, but these changes were absent 3 days later. Volinanserin did not cause any additional changes in grooming and unsupported rearing in naïve mice, consistent with lack of effect on locomotory activity.

We have also tested whether a lower dose of psilocybin would induce similar anti-nociceptive effects. Male mice underwent SNI surgery and, when static mechanical hypersensitivity was fully developed (day 12), they received a single intra-peritoneal (i.p.) injection of psilocybin (0.3 mg/kg) or saline control (Fig. 2A). An increase in HTR was observed in the psilocybin treatment groups compared to saline control (Fig. 2B) further confirming central nervous system exposure to the drug. Mechanical hypersensitivity was reduced (Maximum Possible Effect, MPE = 18%) in mice treated with psilocybin with the effect lasting up to day 28 after injection (test day 40) (Fig. 2C). Moreover, preliminary observations were that affective behaviours that characterise dynamic mechanical hypersensitivity to light brush stroke after psilocybin treatment were reduced (Fig. 2D), but that no effect was observed on the licking/biting response to a cold stimulus (Fig. 2E). The single dose of psilocybin (0.3 mg/kg) had no effect on locomotor performance in SNI mice (Fig Supplementary Fig 2B). Taken together, we observed a number of anti-nociceptive effects of psilocybin in the SNI model, consistent with the potential alleviative effects in the human pain experience. We next tested whether repeated injection of psilocybin (0.3 mg/kg) could amplify the anti-nociceptive effects observed with a single dose. Psilocybin was injected every 7 days for 3 weeks in male mice that had undergone SNI or sham surgery (Fig. 2F). A reduction of mechanical hypersensitivity was observed in SNI mice. Repeated injections of psilocybin (0.3 mg/kg) substantially prolonged and amplified the anti-nociceptive effects for several weeks (Fig. 2G) in comparison to a single dose (Fig. 2C) (%MPE = 62.6 in the same group of mice (SNI) before and after psilocybin injection). No changes in mechanical threshold were observed in sham mice (Fig. 2G). These findings indicate that a repeated low-dose psilocybin regimen can sustain and enhance analgesic efficacy beyond that achieved with a single administration, extending recent preclinical evidence for psychedelic-assisted analgesia and, to our knowledge, representing the first demonstration of this dosing strategy in a neuropathic pain model.

### Higher-dose psilocybin enhances gabapentin effects

The neural mechanisms underlying the effects of psilocybin are not fully understood but are thought to involve the modulation of normal patterns of communication between different areas of the brain. Altered brain functional connectivity in chronic pain patients and mice have been observed[25,26] and the analgesic effect of psychedelic drugs might be due to their capacity to drive neuroplasticity and reset aberrant connections that support chronic pain[27]. Here, we hypothesised that psilocybin may be able to influence pain processing networks in mice beyond the period when alterations in pain behaviours were seen.

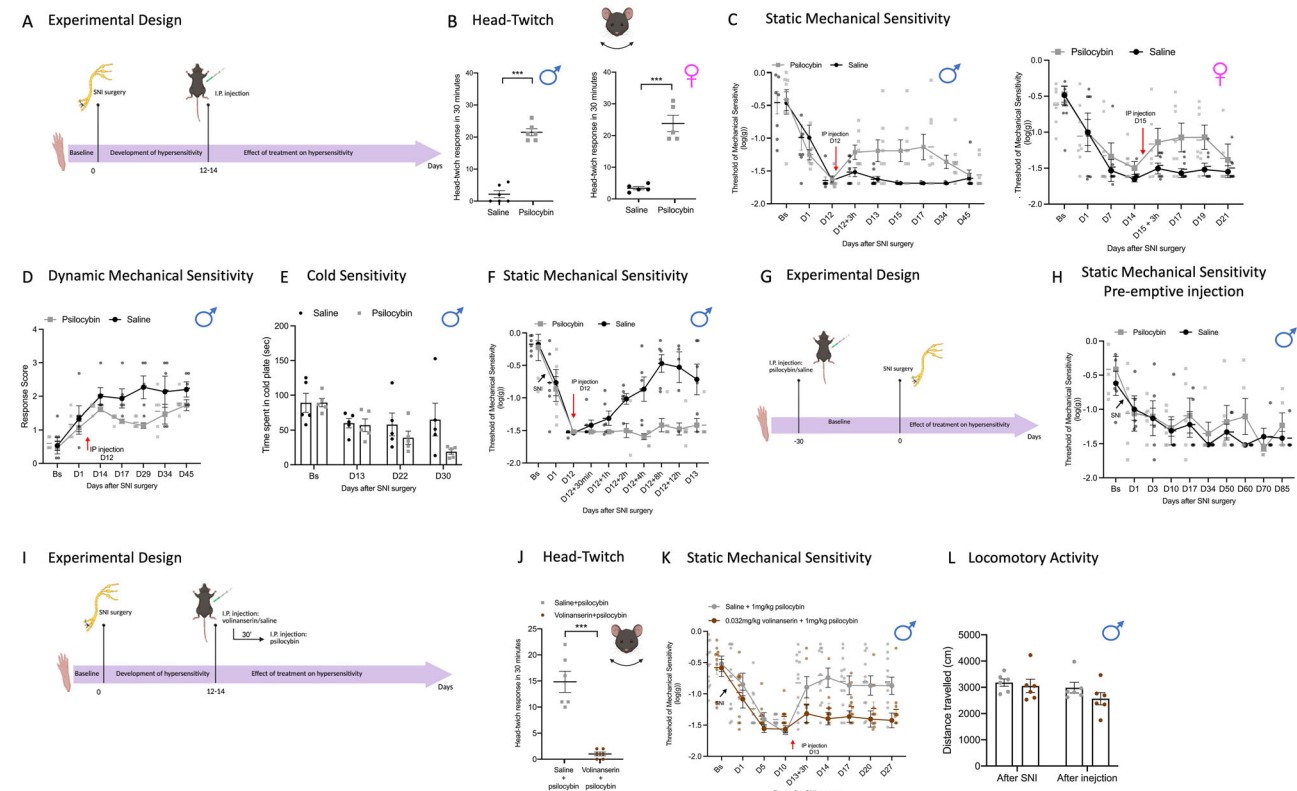

**Fig. 1 | Psilocybin (1 mg/kg) improves hypersensitivity in male and female mice.**
**A** Schematic of experimental design. Behavioural tests were performed before (baseline) and after SNI surgery until the end of experiments. Psilocybin (1 mg/kg) was injected when max sensitivity was fully developed (between day 12 and day 15). **B** Head-twitch response in male and female mice after injection of psilocybin (1 mg/kg) (male mice, n = 6; female mice, $n = 5$) or saline control (male mice, n = 6; female mice, $n = 5$); $P < 0.001$ unpaired independent sample $t$-test. **C** Static mechanical threshold of mice assessed using calibrated von Frey filaments before (Bs, baseline) and after SNI surgery. At maximum sensitivity day 12 (male) or on day 15 (female), mice received an IP injection of 1 mg/kg psilocybin or saline control (male mice, n = 8/8, two-way repeated-measures mixed-model ANOVA, factor 'treatment' 3 h to D45: $F = 10.05$, $P = 0.007$; female mice, $n = 8/8$, two-way repeated-measures mixed-model ANOVA, factor 'treatment' D15 + 3 h to D21: $F = 12.1$, $P = 0.005$) (on D21 psilocybin $n = 5$ mice). **D** Brush-evoked dynamic hyper-sensitivity before (Bs) and after SNI surgery ($n = 5/5$ mice, two-way repeated-measures mixed-model ANOVA, factor 'treatment' D14–D45: $F = 6.5$, $P = 0.034$). **E** Cold allodynia assessed using Thermal Preference Test before (Bs) and after SNI surgery ($n = 5/5$ mice, two-way repeated-measures mixed-model ANOVA, factor 'treatment' D13 to D30: F = 3.9, P = 0.085, with Bonferroni corrections D22, $P = 0.09$, D30, $P = 0.02$). **F** Acute effect of psilocybin on static mechanical threshold assessed using calibrated von Frey filaments. At maximum sensitivity day 12 mice received an IP injection of 1 mg/kg psilocybin or saline control. Behavioural measures taken at several time points after injections (30 min, 1 h, 2 h, 4 h, 8, 12 h and 24 h). ($n = 6/6$ mice, two-way repeated-measures mixed-model ANOVA, factor 'treatment' 30 min to D13: $F = 18.7$,

$P = 0.002$; with Bonferroni corrections 1 h, $P = 0.085$, 2 h, $P = 0.005$, 4 h, $P = 0.004$, 8 h, $P < 0.001$, 12 h, $P = 0.004$, 24 h, $P = 0.022$). **G** Schematic of experimental design. Von Frey test was performed before (baseline) and after SNI surgery until the end of experiments. Psilocybin (1 mg/kg) was injected 30 days before SNI surgery. **H** Static mechanical threshold of mice before (BS, Baseline) and after SNI surgery (D1–D85). ($n = 5/5$, two-way repeated-measures mixed-model ANOVA, factor 'treatment' Bs to D85: $F = 0.64$, $P = 0.45$). **I** Schematic of experimental design. Von Frey test was performed before (baseline) and after SNI surgery until the end of experiments. On day13 mice received an injection of 0.032 mg/kg of volinanserin or saline control; 30 min later all mice received an IP injection of 1 mg/kg of psilocybin. **J** Head-twitch response in male mice after injection of psilocybin (1 mg/kg) and volinanserin (0.032 mg/kg) ($n = 6/6$ mice, $P < 0.001$ unpaired independent sample $t$-test). **K** Static mechanical threshold of mice before (BS, Baseline) and after SNI surgery (D1–D27). ($n = 12/12$ mice, two-way repeated-measures mixed-model ANOVA, factor 'treatment' D13 + 3 h to D27: $F = 11.9$, $P = 0.002$).
**L** Locomotor activity (distance travelled) after SNI surgery and subsequent injection of saline or psilocybin. Bar graph displays the total distance travelled (cm) by animals following SNI ("After SNI") and after acute injection ("After injection") of either saline + psilocybin (grey circles) or volinanserin + psilocybin (brown circles). Data are presented as mean ± SEM, with individual data points shown ($n = 6/6$ mice). Data are expressed as mean ± SEM throughout. Red arrows represent psilocybin injection. SNI, Spared Nerve Injury; IP, Intra-peritoneal. **A, B, G, I, J** created in https://BioRender.com/76ibtsy.

To test this hypothesis, we conducted two complementary experiments examining psilocybin's interaction with gabapentin at different timepoints. In the first experiment, gabapentin (50 mg/kg i.p.) was administered during the peak anti-nociceptive effect of psilocybin (1 mg/kg i.p.) (Fig. 3A). This experiment revealed that co-administration of psilocybin and gabapentin produced significantly enhanced and prolonged analgesia compared to gabapentin alone, demonstrating an additive or synergistic interaction during the acute window of psilocybin's direct pharmacological effects (Fig. 3B). We observed significant main effects of both time and treatment on post-SNI mechanical sensitivity, with a significant interaction between these factors indicating a potential additive and/or synergistic effect of treatment over the disease time course. To further investigate this observation, we have carried out post-hoc tests with Bonferroni corrections at time day 30, 33,

$33 + 30$ min, $33 + 60$ min, $33 + 90$ min. This analysis revealed reduced mechanical hypersensitivity in the psilocybin treated group before the administration of gabapentin ($P = 0.004$), no difference between the two treatments during the first 60 min of gabapentin administration, but a significant reduction in mechanical hypersensitivity in the psilocybin + gabapentin treated group at 90 min in comparison to the control ($P = 0.018$). This observation suggests that psilocybin here exerts an additive and/or synergistic effect on gabapentin-induced anti-nociceptive effects.

Importantly, in a second experiment, gabapentin was administered at day 55 after surgery, a timepoint at which the direct anti-nociceptive effect of psilocybin was no longer measurable (Fig. 3C). In mice previously treated with psilocybin, gabapentin produced a dramatic and sustained anti-nociceptive effect lasting from 2 to 96 h, in stark contrast to the markedly

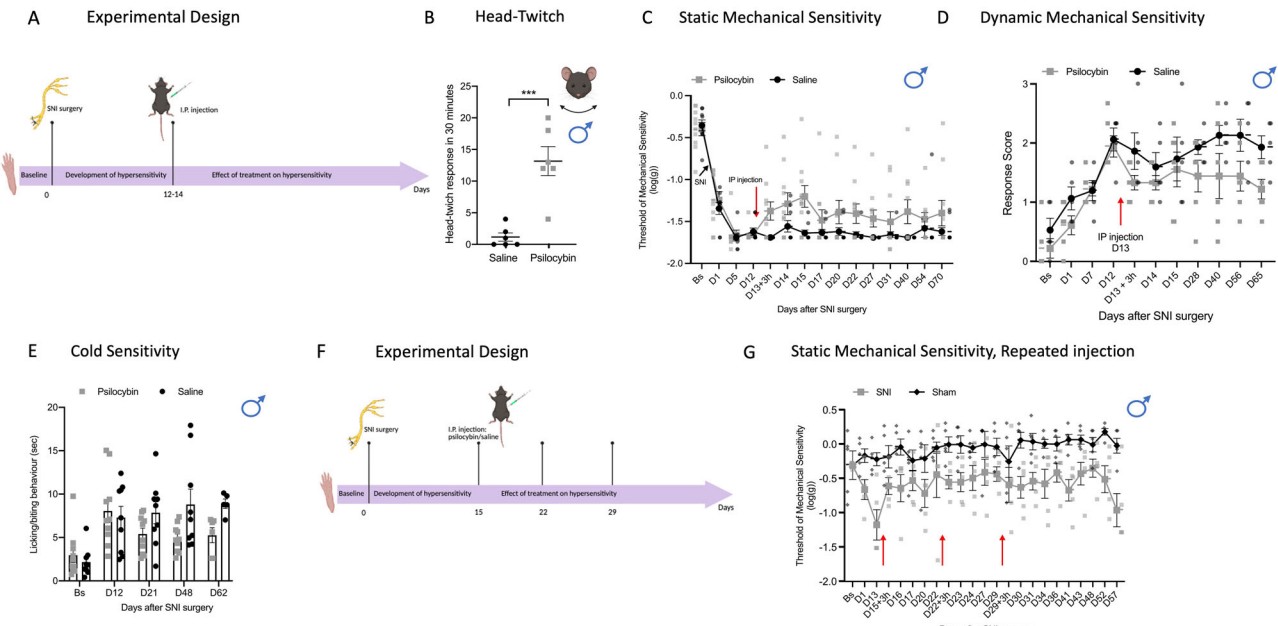

**Fig. 2 | Psilocybin (0.3 mg/kg) attenuates pain-like behaviour in male mice after peripheral nerve injury. A** Schematic of timeline of the experiments. Behavioural tests were performed before (baseline) and after SNI surgery until the end of experiments. Psilocybin (0.3 mg/kg) was injected when max sensitivity was fully developed (between day 12 and day 15). **B** Head-twitch response after injection of psilocybin (0.3 mg/kg) ($n = 6$) or saline control ($n = 6$). $P = 0.001$, unpaired independent sample $t$-test. **C** Static mechanical threshold of mice assessed using calibrated von Frey filaments before (Bs, baseline) and after SNI surgery (D1–D40) ($n = 10/9$, two-way repeated-measures mixed-model ANOVA, factor 'treatment' D13 + 3 h to D54: $F = 10.7$, $P = 0.004$). **D** Brush-evoked dynamic hypersensitivity before (Bs) and after SNI surgery ($n = 6/5$, two-way repeated-measures mixed-model ANOVA, factor 'treatment' D13–D65: $F = 7.2$, $P = 0.025$). **E** Cold

allodynia assessed using acetone drop applied to the hindpaw ipsilateral to the injury (left) before (Bs) and after SNI surgery ($n = 10/9$, two-way repeated-measures mixed-model ANOVA, factor 'treatment' D21 to D62: $F = 2.8$, $P = 0.13$).
**F** Schematic of timeline of the experiments. Behavioural tests were performed before (baseline) and after SNI surgery until the end of experiments. On day 15, 22 and 29 after SNI, all mice received an IP injection of psilocybin (0.3 mg/kg) (red arrows). **G** Static mechanical threshold of mice assessed using calibrated von Frey filaments before (BS, baseline) and after SNI or SHAM surgery. ($n = 5/5$, two-way repeated-measures mixed-model ANOVA, factor 'treatment' D15 + 3 h to D57: $F = 13.7$, $P = 0.006$). Data are expressed as mean ± SEM throughout. Red arrows represent psilocybin injection. SNI, Spared Nerve Injury; IP, Intra-peritoneal. **A, F** created in https://BioRender.com/76ibtsy.

---

attenuated and shorter-duration response observed in mice treated with saline vehicle (Fig. 3D). These findings are consistent with administration of a single dose of psilocybin producing lasting changes to pain-processing networks that persistently enhance the efficacy of established analgesic medications. Critically, this sustained enhancement of gabapentin activity suggests a fundamental restructuring of pain-processing networks that extends well beyond psilocybin's acute pharmacological window and points to a novel therapeutic strategy for chronic pain management. Collectively, these data demonstrate that psilocybin exhibits (i) acute anti-nociceptive effects that enhance concurrent analgesic responses and (ii) sustained effects that persistently modify pain-processing networks and amplify analgesic efficacy weeks after initial administration.

## Discussion
Developing safer, effective treatments for chronic pain has proven challenging. Here, we demonstrate that psilocybin can reduce neuropathic pain-like behaviour in male mice for up to 30 days, that this reduction in pain behaviour can be amplified by repeated treatment with low-dose psilocybin, and critically, that psilocybin, both acutely and in a sustained manner, can potentiate the effect of gabapentin, a standard treatment for neuropathic pain in humans. Recent research has reported no general differences between males and females in response to psilocybin in rodent models of inflammatory and neuropathic pain[9,12] and here we report that psilocybin also reduces mechanical sensitivity in female mice in the SNI model of neuropathic pain. A single dose of psilocybin also had a persistent effect in female mice, but a fuller characterization of psilocybin anti-nociceptive effects in female animals is warranted to determine if sex differences, such as the duration of effect on mechanical sensitivity we report here, extend to other measures.

Notably, a single recent study reported that psilocybin at doses up to 10 mg/kg lacked analgesic effects in several mouse pain models[28]; this is in contrast to other recent reports of anti-nociceptive effects in models of chemotherapy-induced peripheral neuropathy (CIPN)[29,30] and inflammatory pain[12] (reviewed in ref. [15]) and our data in the SNI model here (also see 30). Our experimental design facilitated detection of sustained mechanism of pain reduction that might not be captured in acute paradigms and include an initial consideration of sex differences. Consistent with these findings, recent work has demonstrated that single-dose psilocybin rapidly and sustainably relieves pain-like behaviours while simultaneously alleviating co-occurring anxiodepressive symptoms across multiple chronic pain models, with these effects mediated through normalized activity in the anterior cingulate cortex via partial agonism at both 5-HT$_{2A}$ and 5-HT$_{1A}$ receptors[9]. Moreover, direct application od psilocin into the anterior cingulate cortex reversed pain-like behaviour[9]. We further this work with the first report that psilocybin can potentiate the analgesic effects of extant medications.

Interestingly, we also shown that a pre-emptive injection of psilocybin administered before SNI surgery was unable to prevent the development of mechanical hypersensitivity. This finding provides important insight into the temporal dynamics of psilocybin's therapeutic mechanism and suggests that the drug's beneficial effects on pain processing depend critically on the presence of established maladaptive pain networks. Rather than functioning as a protective agent against the initial injury-induced pathological pain states, psilocybin appears to operate as a therapeutic agent capable of modulating networks that have already become dysfunctional. This distinction is mechanistically important: the acute surgical trauma and immediate inflammatory cascade triggered by SNI surgery generate rapid pain-processing changes and network reorganization[31] that likely outpace

**A** Experimental Design

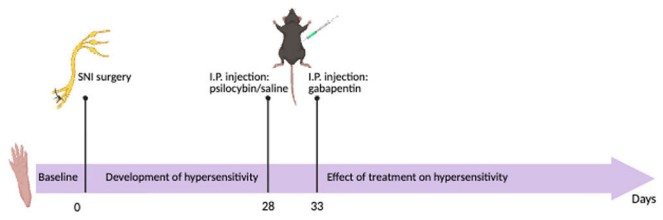

**B** Static Mechanical Sensitivity

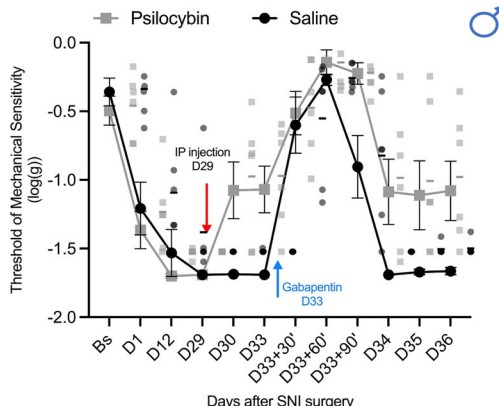

**C** Experimental Design

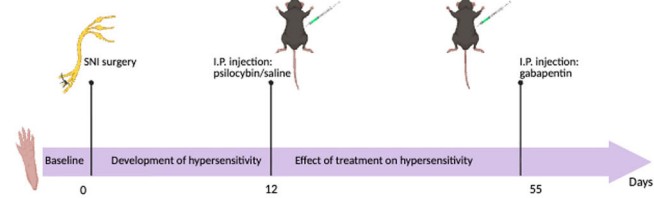

**D** Static Mechanical Sensitivity

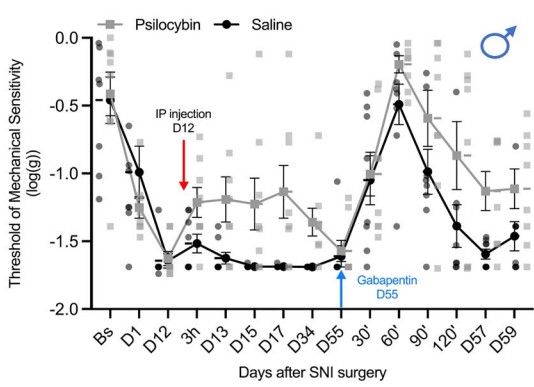

**Fig. 3 | Psilocybin potentiates the effect of gabapentin in male mice. A** Schematic of timeline of the experiments. Von Frey test was performed before (baseline) and after SNI surgery until the end of experiments. Psilocybin (1 mg/kg) or saline control were injected IP on day 29. On day 33, all mice received an IP injection of gabapentin (50 mg/kg). **B** Static mechanical threshold of mice before (BS, Baseline) and after SNI surgery (D1–D36). After injection of gabapentin, von Frey test was performed at 30', 60' and 90', then daily until the end of the experiment (Day 36). ($n = 6/6$, two-way repeated-measures mixed-model ANOVA, factor 'Treatment' $F = 10.78$, $P = 0.008$; Within subject, Factor 'Time', $F = 37.3$, $P < 0.001$; Factor 'Time x

Treatment', $F = 2.78$, $P = 0.039$). **C** Schematic of timeline of the experiments. Von Frey test was performed before (baseline) and after SNI surgery until the end of experiments. Psilocybin (1 mg/kg) or saline control were injected when max sensitivity was fully developed (day 12). **D**, Gabapentin (50 mg/kg) was injected i.p. 55 days after SNI surgery (blue arrow). ($n = 8/8$, two-way repeated-measures mixed-model ANOVA, factor 'treatment '60' to D59: $F_{1,14} = 9.59$, $P = 0.008$). Data are expressed as mean ± SEM throughout. Red arrows represent psilocybin injection. Blue arrow represents injection gabapentin injection. SNI, Spared Nerve Injury; IP, Intra-peritoneal. **A, C** created in https://BioRender.com/76ibtsy.

psilocybin's network-restructuring capacity when the drug is administered before the injury. Conversely, when administered to established chronic pain states, psilocybin can effectively remodel the consolidated, maladaptive connectivity patterns that have become established through repeated nociceptive signalling and network-level changes.

Human studies have confirmed that 5-HT$_{2A}$R activation is a likely key driver of the profound changes in perception and consciousness caused by psilocybin, but the role of 5-HT$_{2A}$Rs in mediating clinical efficacy remains to be determined fully. Rodents studies have established that 5-HT$_{2A}$R activation is necessary to drive some changes considered to be proxies of therapeutic effects of psilocybin[32], and our data demonstrates that volinanserin, a selective 5-HT$_{2A}$R antagonist, can prevent the full anti-nociceptive effects of psilocybin. By contrast, one major earlier behavioural study reported that ketanserin, a 5-HT$_{2A/2C}$R antagonist, did not influence psilocybin's antidepressant-like behaviours, suggesting that the receptor mechanisms underlying different therapeutic effect of psilocybin may be distinct[33,34]. Other studies using ketanserin indicate an, at least partial, involvement of 5-HT$_{2A}$Rs in schizophrenia-like psychosis[35] and cognitive functions[36]. Psilocin also has appreciable affinity at other 5-HTRs, including 5-HT$_{1A}$R. Recent research using volinanserin has

shown complete blockade of psilocybin's anti-nociceptive effects in chemotherapy-induced peripheral neuropathy (CIPN) models[29,37]. In these studies, volinanserin pretreatment (0.05 mg/kg) completely blocked the anti-nociceptive effects of both psilocybin and DOI, indicating that 5-HT$_{2A}$R activation is necessary for psilocybin's analgesic properties in neuropathic pain models. These findings support that the analgesic properties of psilocybin require 5-HT$_{2A}$R activation. We show that volinanserin does not fully block psilocybin anti-nociceptive effects in SNI model; these data suggest that different dosing regimens may have to be explored or that other mechanisms may also contribute. In this regard, a significant body of evidence points to BDNF release being an important downstream consequence of psilocybin administration[38,39]. Neurons in both the frontal cortex and dorsal horn of the spinal cord have been shown to respond to psilocybin by augmentation of BDNF signalling[38]. Psilocybin induces a rapid and persistent growth of dendritic spines of medial prefrontal cortex neurons[19] which may contribute to the increased excitability of neurons and a reduction in the chronic pain behaviour reported[25]. Future investigations are therefore required to establish whether the anti-nociceptive effect of psilocybin could also be linked to the action of BDNF released in the cortex and dorsal horn and whether

BDNF-mediated neuroplasticity contributes to the sustained enhancement of gabapentin efficacy observed in our studies.

Gabapentin is a drug widely used in clinical practice to treat neuropathic pain; however, not all the people with neuropathic pain achieve adequate pain relief following administration[40]. Moreover, gabapentin use is also associated with side effects[40] and a risk of addiction[41]. Here we investigated the temporal dynamics and persistence of psilocybin's ability to enhance gabapentin-mediated anti-nociceptive efficacy. The administration of gabapentin during the peak anti-nociceptive effect of psilocybin revealed a marked enhancement of gabapentin's analgesic efficacy, demonstrating that co-administration produces significantly prolonged pain relief compared to gabapentin alone. This acute effect indicates that psilocybin's effects can substantially amplify the therapeutic action of conventional analgesics.

More strikingly, we have shown a dramatic and sustained enhancement of gabapentin's anti-nociceptive efficacy when administered weeks after psilocybin treatment, at a time when psilocybin's direct anti-nociceptive effects were no longer detectable. This sustained potentiation represents a fundamental departure from traditional pharmacokinetically-dependent drug-drug interactions and instead may reflects a sustained mechanism by which a pharmacodynamic drug-drug interaction can occur. One potential mechanism here is that psilocybin may induce persistent neuroplastic changes[19,42] that alter pain processing networks; this may create a neurobiological environment more conducive to gabapentin's therapeutic actions. Recent evidence demonstrates that psilocybin-induced alterations in anterior cingulate cortex activity and connectivity establish a neural substrate upon which conventional analgesics can operate more effectively. Several neurobiological mechanisms could explain psilocybin's ability to enhance gabapentin analgesic efficacy weeks after administration. For example, BDNF expression in the rostral ventromedial medulla (RVM) has been shown to be essential for morphine's analgesic effects and can potentiate morphine efficacy at otherwise ineffective doses[43]. The improved efficacy of gabapentin observed following psilocybin treatment strongly justifies investigating whether similar enhancement occurs with other drugs used to target chronic neuropathic pain, such as morphine, amitriptyline and duloxetine[44].

The psilocybin and gabapentin doses used in this study were selected to fall within ranges that reliably engage their primary molecular targets and reverse mechanical hypersensitivity in rodent neuropathic pain models, while avoiding non-specific locomotor impairment or sedation. In particular, our psilocybin regimen is consistent with doses that produce robust 5-HT$_{2A}$ receptor-dependent behavioural effects and long-lasting structural and functional plasticity in medial frontal cortex circuits, including persistent increases in dendritic spine density and excitatory synaptic drive, which have been proposed to underlie durable therapeutic actions of the drug[19]. Gabapentin dosing was chosen on the basis of extensive pharmacological characterization of the SNI and related nerve-injury models, where comparable regimens reliably reverse mechanical allodynia and affective components of neuropathic pain without motor confounds, thereby ensuring that our combination paradigm is anchored in clinically relevant exposures[45,46].

Sample sizes for each experiment were decided a priori on the basis of previous work using the SNI model and related psilocybin–pain studies, together with our pilot data on variability in von Frey thresholds and are comparable to those commonly employed in neuropathic pain research; exact n values are reported in each figure legend. Because this study was designed as an exploratory preclinical investigation of sustained and combinatorial effects, group sizes were chosen to balance the ability to detect biologically meaningful changes with the need to minimise animal use, in line with the ARRIVE 2.0 recommendations on sample-size justification for feasibility studies. Mechanistically, combining these dosing and sampling strategies with pharmacological blockade of 5-HT$_{2A}$ receptors allows us to propose that psilocybin acts via 5-HT$_{2A}$ receptor-driven network within anterior cingulate and spinal pain circuits, providing a plausible substrate both for its direct anti-nociceptive effects and for the delayed, sustained enhancement of gabapentin efficacy observed in our study.

Together, these data provide the first preclinical demonstration that psilocybin could be an effective tool for the management of chronic pain due to nerve injury across sexes and offer a new therapeutic adjunct for the control of chronic pain. The potentiation of gabapentin efficacy weeks after psilocybin administration represents a finding that warrants immediate investigation with morphine and other analgesics.

This approach offers a paradigm shift in translatable chronic pain management: rather than developing entirely new drugs, a single dose of psilocybin could unlock the therapeutic potential of existing, well-characterised analgesics in treatment-resistant patients. The convergent evidence from 5-HT$_{2A}$R involvement and potential network connectivity changes provides a strong mechanistic foundation to address critical clinical needs in pain medicine while establishing psilocybin as a network-restructuring agent that enhances the efficacy of standard-of-care treatments.

## Methods

### Experimental design

This study was designed to evaluate the effect of psilocybin on pain sensitivity. In all experiments, mice were randomly assigned into treatment groups to ensure unbiased allocation. Confounders were not controlled. Allocation details were concealed, and cages were coded so that the experimenter performing the behavioural assessments and analyses remained fully blind to treatment conditions throughout the study. Blinding was maintained during drug preparation, testing, and statistical evaluation to minimize potential bias. The exact number of mice per group is specified in the respective figure legends.

### Animals

All experimental procedures were conducted in accordance with institutional and national guidelines for the care and use of laboratory animals, were approved by the relevant ethical review board, and are reported in compliance with the ARRIVE (Animal Research: Reporting of In Vivo Experiments) guidelines[47]. All procedures were performed in accordance with the UK Animals (Scientific Procedures) Act, 1986, and the principles of the 3Rs (Replacement, Reduction, and Refinement). We have complied with all relevant ethical regulations for animal use. Every effort was made to minimize potential suffering, and the number of animals used was reduced to the minimum required to ensure sufficient statistical power. Sample size calculations were informed by variance estimates and effect sizes obtained from our previous behavioural assays, ensuring that group sizes were adequate for reliable detection of treatment effects without unnecessary animal use. A total number of 157 mice have been used for this study. No criteria were set for including and excluding animals.

Adult male and female C57BL/6 J mice (8–12 weeks old) were obtained from Charles River Laboratories (UK) and acclimatized to the housing facility for at least one week prior to experimentation. Animals were group-housed in individually ventilated cages (maximum of five per cage) under standard laboratory conditions, with controlled ambient temperature (20 ± 1 °C) and humidity (55 ± 5%). A 12:12 h light–dark cycle was maintained (lights on at 07:30 a.m.), and animals had free access to food and water throughout the study. Environmental enrichment (nesting material and shelters) was provided in all cages to promote welfare.

All experimental protocols were approved by the institutional Animal Welfare and Ethical Review Body (AWERB) and were carried out under UK Home Office Project Licence PPL PP9720547.

### Mouse model of neuropathic pain: spared nerve injury (SNI)

The SNI surgery was performed as described by Decosterd & Woolf,[16]. Briefly, under isoflurane anaesthesia, the skin on the lateral surface of the thigh was incised, and a section made directly though the biceps femoris muscle, exposing the sciatic nerve and its three terminal branches: the sural, the common peroneal, and the tibial nerves. The common peroneal and the tibial nerves were tightly ligated with 5–0 silk suture and sectioned distal to the ligation. Great care was taken to avoid any contact

with the spared sural nerve. Complete haemostasis was confirmed, and the wound was sutured.

## Drugs

Psilocybin (COMP360, a proprietary formulation of synthetic psilocybin) was provided by Compass Pathfinder (a subsidiary of Compass Pathways) and was dissolved in saline. Control mice received equal volume of saline.

Gabapentin was purchased from Sigma and was dissolved in saline.

Volinanserin was purchased from Merck and was dissolved in saline.

## Behavioural testing

**Von Frey filament test for static mechanical sensitivity.** For the assessment of mechanical sensitivity, the von Frey filament test was used. Mice were placed in Plexiglas chambers, located on an elevated grid, and allowed to habituate for at least 1 h. After this time, the plantar surface of the paw was stimulated with a series of calibrated von Frey monofilaments[48], mechanical sensitivity threshold was determined using the up-down-method[49]. The data were expressed as log of the mean of 50% pain threshold ± SEM.

## Brush test for dynamic mechanical sensitivity

Dynamic allodynia was tested by light stroking (velocity ~2 cm/s) of the external lateral side of the injured hind paw in the direction from heel to toe using a paintbrush using a protocol adapted from Duan and colleagues[50]. Mice were placed in Plexiglas chambers located on an elevated grid, and allowed to habituate for at least 1 h. Observed responses were scored: 0, no response or moving the stimulated paw; 1, single withdrawal, flick or stamp of the stimulated paw; 2, multiple withdrawals of the stimulated paw in rapid succession; 3, licking of the plantar surface or continued elevation/withdrawal of the stimulated paw. The stimulation was repeated three times at intervals of at least 3 min and the average scores was obtained for each mouse.

## Acetone test for cold sensitivity

For assessment of cold sensitivity, the acetone test was used[48]. Mice were placed in Plexiglas chamber located on an elevated grid for 1 h and then a drop ( ~ 50 ml) of acetone was applied to the external lateral side of the injured hind paw. Total time licking/biting of the hid paw was recorded for 20 s.

## Thermal place preference

The development of cold allodynia was analysed using the thermal place preference (TPP) apparatus (Ugo Basile, Cat. No. 35250), which consist of two cylinders connected by a narrow center walkway. Briefly, mice were initially allowed to explore the apparatus for 5 min with both plates at room temperature 23–25 °C. At the beginning of the test, the temperature of one of the plates was reduced to 15 °C and the amount of time spent in each chamber and the number of transitions between chambers were manually recorded during 10 min.

## Rotarod test

In this study we used an accelerating rotarod apparatus with a 3 cm diameter rod starting at an initial rotation of 4 rpm and slowly accelerating to 40 rpm over 100 s. Mice were expected to walk at the speed of rod rotation to keep from falling. Mice were not tested at baseline to minimize the number of tests on the apparatus. Time taken to fall from the rod was recorded[51].

## Fecal pellet output

Mice were placed in Plexiglas chamber located on an elevated grid for 1 h, after this time the number of fecal pellets were counted for each mouse.

## Head-twitch response

After the drug or saline was administered, the mice were placed in individual Plexiglas chambers and movement recorded using a digital camera. The videos for each mouse were subsequently analyzed to count the number of head twitches, which are characterised by rapid, involuntary movements of the head. The number of head twitches was counted over a 30 min period.

## Spontaneous behaviour

Spontaneous behaviour was assessed using the EthoVision XT video tracking system. Each animal was placed individually in the open-field arena, and behaviour was recorded for the duration of the test session (10 min). Automated tracking was used to quantify total distance travelled (cm) as a measure of locomotor activity. In addition, ethological parameters including unsupported rearing and grooming episodes were coded and analysed, providing indices of exploratory behaviour and self-maintenance, respectively. All recordings were performed under consistent lighting and environmental conditions to minimize variability.

## Open field test

The open field test (OFT) was used to analyze the effect of psilocybin on locomotor activity (measured as the total distance travelled and the frequency of entering the center zone). Mice were tested 24 h post-injection with 1 mg/kg of psilocybin or saline vehicle. Mice were placed in the center of a circular plastic box (33 cm × 33 cm) and allowed to move freely. The total distance travelled and the frequency of entering the center of the arena were recorded over the next 5 min; the EthoVision Track System was used to record and analyze the behaviour.

## Statistics and reproducibility

All statistical tests were performed using the IBM SPSS Statistic Programme (version 27), and $P < 0.05$ was considered statistically significant. Data are means ± s.e.m. and independent experiment unit (n) is animals. Data of von Frey filaments test was log transformed to ensure a normal distribution[52,53].

Differences in sensitivity were assessed using repeated-measures mixed-model ANOVA, with 'time' specified as a within-subject factor and 'treatment' as a between-subject factor. Following a significant interaction or main effect, pairwise comparisons were performed using Bonferroni correction to control for multiple testing. Where comparisons across all group means were required, a Tukey's post-hoc test was applied. Sample sizes varied depending on the experiment and ranged from $n = 5$ to $n = 12$ per group. Exact sample sizes and relevant statistical details are provided in the figure legends. All animals used in the study have been included in the analysis. Detailed two-way ANOVA statistics (P values, observed power and partial $\eta^2$) for all panels are reported in Supplementary Fig 5.

The MPE (Maximum Possible Effect) was calculated according to the formula:

$$\%MPE = \frac{100 \times [\log(\text{drug induced threshold}) - \log(\text{vehicle induced threshold})]}{[\log(0.6) - \log(\text{vehicle induced threshold})]}$$

where $\log(0.6)$ is our maximum von Frey's force applied.

## Reporting summary

Further information on research design is available in the Nature Portfolio Reporting Summary linked to this article.

## Data availability

Numerical source data for all graphs in the manuscript can be found in supplementary data 1 file. Data supporting the conclusions of this study can be accessed through the University of Reading Research Data Archive; https://doi.org/10.17864/1947.001510.

## Materials availability

All data associated with this study are available in the main text or the supplementary materials.

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

## Acknowledgements

We gratefully acknowledge the staff of the University of Reading Bioresource Unit for their invaluable support, and Dr Harvey Roweth for his expert assistance with the figure graphics and his insightful contributions to the manuscript. Funding: This work was funded in whole or in part by the Academy of Medical Sciences Springboard SBF008\1092 to M.M. and the University of Reading Strategic PGR Studentships (T.A. and D.L.). This funding included industrial partnership award grant in collaboration with Compass Pathfinder Limited (a subsidiary of Compass Pathways) (G.G., T.A., D.L., R.L. and M.A.).

## Author contributions

T.A., M.A., S.P.H., G.J.S. and M.M. conceived and designed the experiments; T.A., D.A.R., D.L. and M.M. performed the experiments; F.T. and M.M. analysed the data; M.M., R.L., G.G. and G.J.S. supervised the students working on the project; T.A., M.A., S.P.H., G.J.S. and M.M. drafted the paper and M.M. wrote the manuscript. All authors revised and edited the manuscript.

## Competing interests

The authors declare no competing interests.
