## [Transparent Peer Review file · Communications Biology]

Psilocybin ameliorates neuropathic pain-like behaviour in mice and facilitates the gabapentin-mediated analgesia

Corresponding Author: Dr Maria Maiarú

Responses to the Reviewers

Reviewer #1 (Remarks to the Author):

The study investigates the analgesic potential of psilocybin in a murine neuropathic pain model and its synergistic interaction with gabapentin. While the topic holds significant translational relevance given the need for non-addictive chronic pain therapies, several concerns warrant clarification and refinement.

We would like to thank the Reviewer for taking the time to review our revised manuscript and for recognising the significant translational relevance of our work. We have now further refined and clarified the manuscript with a number of additional (including mechanistic) experiments, as described fully below.

1. BDNF actions in spinal cord after peripheral nerve injury contribute to the increased excitability of spinal neurons that underlies neuropathic pain. However, the authors hypothesize that BDNF may mediate psilocybin's analgesic effects.

We have performed further mechanistic studies that now demonstrate that the anti-nociceptive effects of psilocybin, as well as the effects on head-twitch, are mediated by 5-HT_{2A} receptors. We originally tested the effects 5HT_{2A} receptor antagonist volinanserin at 2mg/kg. In response to other Reviewer points regarding concerns about sedative effects of this dose of volinanserin, we now conducted experiments using volinanserin at 0.032 mg/kg. These experiments show that the anti-nociceptive effects of psilocybin are indeed mediated by 5-HT_{2A} receptors (see new figure 2, panel g-h-i). Previously, we also had some concerns regarding the full solubility of volinanserin at 2 mg/kg and we believe that this may explain the discrepancy. Moreover, we also show in new experiments that 0.032 mg/kg volinanserin has no adverse effects on locomotor activity or any spontaneous behaviours measured (including unsupported rearing and grooming) (see figure S5). These additional data directly address molecular pathways, which we agree were a deficit in the original submission. Of further interest was that our data show that volinanserin does not fully block psilocybin anti-nociceptive effects in the SNI model; these data suggest that other mechanisms may also contribute. In our revised discussion we speculate on additional mechanisms (see pages 4 and 5). Whilst not the focus of the present study, it remains possible that neuronal BDNF may contribute to the long-term actions of psilocybin. *Fatt et al., 2025* in *Science* clearly show that the major descending inhibitory pathway from the brain releases BDNF and these neurons contribute to morphine analgesia. Overall, our revised manuscript offers more nuanced reasoning for investigating 5-HT_{2A} receptor involvement and more discussion of clinical significance, adding interpretive context for chronic pain and comorbidities.

2. Psilocybin has the analgesic effect in male and female mice. However, the actions of BDNF regulating neuropathic pain are observed in male rodents but not in female rodents. (Smith PA. BDNF in Neuropathic Pain; the Culprit that Cannot be Apprehended. *Neuroscience*. 2024 Apr 5;543:49-64.)

In the cited paper, the authors investigate microglial BDNF, which is not something we examined in our study. This is partly because there is currently no evidence suggesting that psilocybin has effects on microglia. There is also some contradictory evidence regarding any sex differences in BDNF responses. For example, *Fatt et al., 2025* showed that the descending pain BDNF+ inhibitory pathways from the brain operate in a similar fashion in male and female mice.

3. The study primarily focuses on behavioural phenotypes without exploring the underlying mechanisms. No direct experimental evidence is provided to support these claims. Mechanistic studies would significantly strengthen the manuscript and provide a more comprehensive understanding of psilocybin's action.

As above, we now confirm that volinanserin can prevent the full anti-nociceptive effect of psilocybin and, according, discuss the role of 5-HT_{2A} receptors (see pages 4 and 5). We concur that these mechanistic studies now significantly strengthen the manuscript and provide a more comprehensive understanding of psilocybin's action.

4. The claim that psilocybin potentiates gabapentin's analgesic effect is intriguing. However, it is not preferable to examine the synergistic effect of gabapentin and psilocybin after the mice have been used to examine the analgesic effect of psilocybin. Moreover, the analgesic effect of gabapentin shown in Fig. 2h was stronger than previously reported.

We demonstrate for the first time that psilocybin potentiates the analgesic effects of gabapentin. To properly assess this enhanced effect, it is crucial that gabapentin be administered after psilocybin, as the timing of drug administration plays a key role in the observed interaction. This sequencing allows us to evaluate how psilocybin enhances gabapentin's pain-relieving properties. Moreover, others have previously shown a pronounced reduction of mechanical hypersensitivity in rodents after systemic administration of gabapentin (see DOI: 10.1016/S0304-3959(02)00039-8)

5. The writing of the article lacks thoroughness, as there are typographical errors. On line 146, "HRT" should be corrected to "HTR." On line 59, "C56BL/6" should be changed to "C57BL/6." On line 99, the article states "1 mg/kg," but on lines 91 and 101, it is written as "0.3mg/kg" without a space. Additionally, the superscripts for references are inconsistently placed, with some appearing inside punctuation marks and others outside.

We thank the reviewer for pointing out these typographical errors and they have been corrected.

In summary, this study presents interesting behavioral findings on psilocybin's analgesic effects. The authors found that "Psilocybin facilitates gabapentin-mediated analgesia," but this aspect requires further reinforcement. And the lack of mechanistic exploration limits its impact. Addressing these issues would significantly enhance the manuscript's scientific contribution.

We would like to thank the reviewer for acknowledging that our study 'presents interesting behavioural findings.' As above, we now demonstrate an underlying mechanism. We believe that this adds depth to our findings and strengthens this manuscript as an important first description of psilocybin effects in any model of neuropathy (and further the important effects on gabapentin responses therein).

Reviewer #2 (Remarks to the Author):

The authors have satisfactorily addressed some of my previous comments. However, after reviewing the revised manuscript, I have several concerns that lead me to conclude that this article is not suitable for publication in a top-tier journal such as Nature Communications.

Main Concerns:

1. The number of mice per experimental group is low and varies across experimental conditions, ranging from up to 10 (Fig. 1c) to only 5 (Fig. 4). Given the inherent inter-subject variability, this sample size does not appear to provide sufficient statistical power to support the study's conclusions.

We agree absolutely that sufficient statistical power is a pre-requisite, we do however disagree that just because some experiments are n=5 that they are invalid. To address this point fully, we have conducted an analysis of the power associated with each original data set and the new experiments performed during the revision process; these data are included for the reviewer below (we would be happy to add this table to the main paper, if relevant):

Figure	Two-way ANOVA	Observed Power	η^2p
1 c	P=0.004	87%	0.387
1 d	P = 0.025	67%	0.444
1 e	P = 0.13	32%	0.235
1 f	P = 0.006	90%	0.631
2 b male	P=0.007	84%	0.418
2 b female	P=0.005	88%	0.524
2 d	P=0.034	61%	0.45
2 e	P=0.085	41%	0.326
2 f	P=0.002	97%	0.651
2 h	P=0.002	91%	0.35
2 j	P=0.008	82%	0.407

Thus, we now report Observed Power to provide an indication of the probability of detecting true effects given the sample size (where 80% is commonly accepted threshold). We also report partial eta-squared (η^2p) values as a measure of effect size to indicate the proportion of variance explained by each factor, allowing evaluation of the practical significance of the findings beyond P values (where $\eta^2p \geq 0.14$ is considered a large effect). Further, we clarify that difference in sensitivity was assessed using repeated measures mixed model ANOVA (clarified in the figure legends). In all cases, "time" was treated as within-subject factors and "treatment" was treated as between-subject factor.

More specifically, the data presented in (original) Figure 4 show the actions of volinanserin, using a repeated measures with mixed model ANOVA, testing the overall effect of treatment on the pain-like behaviour, with n=5 we observed a statistical power of 92%, which is higher than the commonly accepted threshold of 80% to support the hypothesis

testing output. As elsewhere, these data are now replaced using a lower dose of volinanserin (figure 2, panel h), but the point remains relevant to address the reviewer concern.

However, the reviewer is correct that, there are a few instances where the observed power does not support any strong conclusion. We feel these data regarding the first descriptions of psilocybin on these measures are of interest to the field as they suggest future directions; however, we now caveat these observations as “preliminary” (see page 2). On this basis, we are satisfied that sample sizes used do provide sufficient statistical power to support the study’s main conclusions. We thank the reviewer for guidance here and believe that the is acceptable practice.

2. The manuscript is brief, and much of the data presented in Figs. 1 and 2 could be consolidated into a single figure, as they both explore the effects of two different doses of psilocybin.

The manuscript was originally submitted as a Short Communication, but now has undergone extensive revision to include a much wider range of data. To address this point, we now consolidate our data (including new experiments) into 2 Figures as requested. Some experiments are shown as Supplemental Data.

3. The antinociceptive phenotypes (e.g., Figs. 1c, 1d, and 1e) lack early post-administration time points, spanning from minutes to a few hours after psilocybin treatment.

We appreciate the reviewer’s comment. In Figures 1c and 1d, as well as in Figure 2b, we did include an early post-administration time point, assessing antinociceptive effects at 3 hours post-injection. To further address this point fully, we have now performed additional experiments on mechanical sensitivity, conducting a 24h time-course following psilocybin or saline injection (See Fig 2 f)

4. The time points (days post-treatment) vary across different figures (Figs. 1, 2, 3, and 4), making comparisons between experiments challenging.

We appreciate the reviewer’s concern, however, we argue that each experiment was designed and conducted to address specific questions, and the observed effects remain robust within their respective timelines. Some differences in choosing when to terminate experiments do not, we believe, compromise the scientific validity or main interpretation of the results.

5. The statistical analysis for the time-course data may be incorrect. The authors do not specify whether they used a two-factor repeated measures ANOVA.

We now clarify that we use two-way repeated measures with mixed model ANOVA and are confident that this is a correct analysis to use. This test allowed us to test the between-subjects factor (Treatment) across the within-subject factor (Time) (we further clarify the latter in the Methods section).

6. Similarly, after performing a two-way ANOVA, the authors used a Student’s t-test rather than an appropriate post-hoc test.

We thank the reviewer for this point. We have now reanalysed all data. Following a significant two-way ANOVA, pairwise comparisons were performed with Bonferroni correction to control for multiple testing (see figure 2e). On one occasion, when comparisons across all group means were required, a Tukey’s post-hoc test was applied (see figure S5). This approach ensured rigorous control of Type I error while allowing both targeted pairwise differences and broader group comparisons to be assessed appropriately. These points have now been added to the Methods section (see page 7).

7. The specific time points at which gabapentin’s effects were statistically compared in Fig. 2h are unclear.

We thank the reviewer for pointing this out. In (new) Fig. 2j, statistical comparisons were made between the two experimental groups at individual time points. We analysed differences between the two groups before injection of gabapentin (D55; Pairwise Comparison with Bonferroni Correction, $p=0.72$) and after gabapentin treatment (D59; Pairwise Comparison with Bonferroni Correction, $P=0.02$). The figure and the figure legends have been now amended accordingly to clarify this analysis.

8. The time points assessed in female mice (Fig. 3) differ significantly from those in male mice, making it difficult to evaluate sex-related phenotypes.

We appreciate the reviewer's comment; however, we do not believe that the difference in time points between male and female mice undermines the primary conclusions here. Firstly, our experimental design is to apply test drugs only when maximum sensitivity is reached and there is no *a priori* reason to believe that maximum sensitivity is the same for different sexes. Secondly, we are committed to end animal experiments once the effect has reversed. It was of interest that this was achieved faster in female rats. However, a more extended analysis of this phenomenon is beyond the scope of the present study. We included this one phenotype in response to a reviewer request and choose to determine if psilocybin had any effect on our primary measure, mechanical sensitivity (see further below).

9. Only one phenotype was evaluated in female mice, limiting the study's ability to draw sex-specific conclusions.

We thank the reviewer for this observation. Female mice were included in the study in response to comments received during the first round of revision. Due to time and resource constraints, we were only able to evaluate one phenotype in females at this stage. Nonetheless, we fully acknowledge the importance of assessing sex-specific effects more comprehensively and are committed to expanding this aspect of the work in future studies.

10. There appears to be a psilocybin effect on static mechanical sensitivity on day 21 in male mice, but not in female mice. This further suggests that the study lacks sufficient statistical power to detect reliable sex differences.

Importantly, at each time point, comparisons were made between experimental groups within the same sex. While the time points differ, the data still provide valuable insight into sex-related phenotypes, and we believe these differences do not affect the overall primary scientific interpretation. We disagree that this necessarily "further suggests that the study lacks sufficient statistical power to detect reliable sex differences". As per table above, this study is sufficiently powered.

Overall, we would like to emphasise that (the new Fig. 2 b) is a nice illustration psilocybin produced clear antinociceptive effects on static mechanical sensitivity in female mice, which we regard as an important take-home message of the study. However, we agree that additional work is needed to fully establish the durability of this effect and introduce some discussion on this point.

11. The dose of volinanserin (2 mg/kg) is excessively high. As a 5-HT_{2A} antagonist, volinanserin may significantly impact locomotor activity, which could explain the altered behavior observed in treated mice. Furthermore, using n=5 per group is insufficient to properly assess the effects of two drugs individually or in combination. Notably, the volinanserin experiment lacks a vehicle-vehicle control. Prior research indicates that doses as low as 0.0032 mg/kg are sufficient to suppress psychedelic-induced head twitches (PMID: 35233648).

We appreciate and agree with the reviewer's feedback. We have now repeated experiments with a dose of 0.032 mg/kg volinanserin (the dose reported by the reviewer is not the one used in the cited study) (see figure 2, panel g, h and i). As per other comment above, this dose was also clearly and fully soluble. These experiments show that the anti-nociceptive effects of psilocybin are indeed mediated by 5-HT_{2A} receptors. Moreover, we also show in new experiments that 0.032 mg/kg volinanserin has no negative effects on locomotor activity or any spontaneous behaviours measured (including unsupported rearing and grooming) (see figure S5). Experiments with 0.032 mg/kg volinanserin are sufficiently powered.

We also acknowledge the absence of a vehicle-vehicle control in this particular experiment, which we will address in future studies to strengthen the experimental design, but should point out that the lack of volinanserin vehicle (saline) effects are shown in several other Figures.

12. The post-hoc test used in Fig. 4 is the LSD test, which does not correct for multiple comparisons, raising concerns about statistical rigor.

As above, we have now conducted multiple pairwise comparisons using Bonferroni correction.

13. Sex as an independent variable has not been evaluated using an appropriate statistical test.

We appreciate the reviewer's observation. Upon reviewing the text, we realize that the phrasing was unclear. In this particular study, we are not directly comparing the two sexes as independent variables. Our intention was to assess the effects of the treatments within each sex individually, rather than performing a direct sex-based comparison. We now remove this point to avoid unnecessary confusion.

Minor Concerns:

The psilocybin dose used in Fig. 3 (female mice) is inconsistent, with 0.3 mg/kg listed in the figure legend and 1.0 mg/kg reported in the main text.

We apologize for the inconsistency and have clarified this in the figure legend.

Reviewer #3 (Remarks to the Author):

I am satisfied with the additions to the manuscript and recommend this for publication. This publication finds evidence for anti-nociceptive effects of psilocybin in a preclinical model, something that has not been previously investigated in a rigorous way. The authors have now made their methods and rationale explicitly clear and I believe the hypotheses, data, and data analyses are sound. This is a very great addition to the field and will facilitate the generation of additional hypotheses and move us forward in our thinking!

We would like to thank the reviewer for their constructive comments and kind words. We are very pleased to hear that the previous additions to the manuscript have addressed the concerns raised and that the study is considered a valuable contribution to the field. We greatly appreciate the recognition of the rigor in our methods and the clarity with which we have presented our rationale. We hope that the additional data now added during revision even further strengthens the reviewer's opinion. It is encouraging to know that our work has the potential to inspire new hypotheses and further advance our understanding of psilocybin's anti-nociceptive effects. We are excited to have our study contribute to this important area of research.